# Juglone in Combination with Temozolomide Shows a Promising Epigenetic Therapeutic Effect on the Glioblastoma Cell Line

**DOI:** 10.3390/ijms24086998

**Published:** 2023-04-10

**Authors:** Anna-Maria Barciszewska, Agnieszka Belter, Iwona Gawrońska, Małgorzata Giel-Pietraszuk, Mirosława Z. Naskręt-Barciszewska

**Affiliations:** 1Intraoperative Imaging Unit, Chair and Department of Neurosurgery and Neurotraumatology, Karol Marcinkowski University of Medical Sciences, Przybyszewskiego 49, 60-355 Poznan, Poland; 2Department of Neurosurgery and Neurotraumatology, Heliodor Swiecicki Clinical Hospital, Przybyszewskiego 49, 60-355 Poznan, Poland; 3Institute of Bioorganic Chemistry of the Polish Academy of Sciences, Noskowskiego 12, 61-704 Poznan, Poland

**Keywords:** juglone, TMZ, GBM, DNA methylation, 5-methylcytosine, 8-oxo-eoxyguanosine, epigenetics

## Abstract

Glioblastoma (GBM) is the most common and aggressive primary brain tumor and one of the human malignancies with the highest mortality. Standard approaches for GBM, including gross total resection, radiotherapy, and chemotherapy, cannot destroy all the cancer cells, and despite advances in its treatment, the prognosis for GBM remains poor. The problem is that we still do not understand what triggers GBM. Until now, the most successful chemotherapy with temozolomide for brain gliomas is not effective, and therefore new therapeutic strategies for GBM are needed. We found that juglone (J), which exhibits cytotoxic, anti-proliferative, and anti-invasive effects on various cells, could be a promising agent for GBM therapy. In this paper, we present the effects of juglone alone and in combination with temozolomide on glioblastoma cells. In addition to the analysis of cell viability and the cell cycle, we looked at the epigenetics effects of these compounds on cancer cells. We showed that juglone induces strong oxidative stress, as identified by a high increase in the amount of 8-oxo-dG, and decreases m^5^C in the DNA of cancer cells. In combination with TMZ, juglone modulates the level of both marker compounds. Our results strongly suggest that a combination of juglone and temozolomide can be applied for better GBM treatment.

## 1. Introduction

Glioblastoma multiforme (GBM) is the most malignant primary tumor of the central nervous system (CNS), with limited therapeutic options and poor prognosis. It constitutes more than 50% of malignant tumors and 14.2% of all CNS tumors [1]. An average patient survival of 8–12 months from diagnosis and a 5-year survival rate is less than 5% [2]. GBM is classified with 4th grade of malignancy (WHO 4). It invades the adjacent brain tissue and is characterized by uncontrolled proliferation, active angiogenesis, invasiveness, and the ability to avoid apoptosis. There is ample evidence that GBM is driven not only by genetics, but also by epigenetic aberrations [3,4]. Standard treatment of glioblastoma patients includes surgical resection, followed by radiotherapy and simultaneous chemotherapy with temozolomide (Figure 1), [5].

Temozolomide is an oral alkylating small molecular mass compound (Figure 1) that prolongs the survival of GBM patients when administered during or after radiotherapy [5]. It is used as first-line chemotherapy and shows significant activity against recurrent gliomas [6]. TMZ goes into the CNS due to its lipophilic properties (log P is equal to -2,8), which enhance the effective crossing of the brain blood barrier (BBB). Under physiological conditions, TMZ is metabolized to MTIC and then to a methyl diazonium ion, which reacts with bases of DNA. The main product is 7-methyl guanosine, but O6 methyl guanosine is a minor one (Figure 2). N-methylated bases are repaired with the BER pathway, but O^6^G is removed by the automethylation of methylguanosine methyl transferase [7]. It is obvious that these two DNA repair pathways counteract the DNA damage induced by TMZ [8].

TMZ interferes with tumor development, slowing its growth and spread in the body [9,10]. Its therapeutic applications can be extended beyond high-grade gliomas [11,12,13]. There are still some open questions regarding TMZ efficacy, patient selection, outcomes, and prognosis. TMZ treatment failure has been observed in the vast majority of glioblastoma patients. Changing the dosing regimen did not meet the expectations concerning the higher effectiveness of the treatment [7,14]. The poor prognosis is probably due to high intrinsic resistance to chemo- and radiotherapy and a rapid spread of glioblastoma cells within the brain. If so, there is an urgent need to develop new drugs that would significantly increase the effectiveness of glioblastoma treatment.

Juglone (5-hydroxy-1,4-naphthoquinone) is a bicyclic small natural origin compound (Figure 1) containing an aromatic ring hydroxylated in the five positions and a quinoid ring with oxygen in the 1 and 4 positions (Log P -1.92) [15,16]. It inhibits various enzymes and affects many cellular processes. Juglone belongs to phenolic, antiseptic compounds which show anticancer, antibacterial, antifungal, and antiviral properties [17,18,19]. Juglone’s antitumor and cytotoxic properties have been demonstrated in various cancer cell lines [20,21,22,23,24]. It is a strong inhibitor of peptidyl–propyl cis/trans isomerase Pin1, which arrests the cell cycle of various cancer types [25,26]. It regulates the phosphorylation of Tau, implicating its potential effects in Alzheimer’s disease [27]. Juglone causes cell death, most likely due to the induction of oxidative stress and damage to the cell membrane [28,29]. Juglone also inhibits the activity of RNA polymerase II and other enzymes by inactivating thiol groups. Up to now, little is known about juglone’s effects on invasion and metastasis. Although juglone shows multidirectional effects on cells, the exact mechanism of its action remains unclear [28,30]. However, the most studied pathway includes ROS. It is known that ROS inducing agents are frequently used in clinical trials for different tumors. Juglone reduces tumor growth through various mechanisms, such as cytotoxicity, apoptosis, and angiogenesis, as a result of increasing ROS levels [31]. The study aimed to evaluate the juglone effect alone and in combination with temozolomide on glioblastoma multiforme cells by looking through the epigenetics window. Both juglone and TMZ could significantly retard glioma growth and increase survival time.

Epigenetics provides a new area for explaining cellular pathological processes. It offers a link between genetic and environmental factors that influence disease development and plays a key role in the pathophysiology of various diseases, from neurological disorders to cancer, and other rare diseases, thus leading to the adaptation of conventional therapies and, ultimately, to better outcomes. Epigenetic modulation includes histone modifications, as well as methylation of DNA, and changes in noncoding RNA profiles. The best characterized epigenetic marker is 5-methylcytosine (m^5^C) in DNA [32,33]. It occurs not only at CpG islands in the proximal promoter region but throughout the whole human genome. DNA methylation regulates gene expression, genomic imprinting, and genome stability. We already showed that global m^5^C contents in DNA could be a diagnostic marker of the malignancy of brain tumors and the severity of other diseases [34,35,36]. We also discovered that temozolomide, although not a substrate for DNMT1, affects the level of methylated cytosine (m^5^C) in DNA. Temozolomide is best known as an alkylating agent of the O6 position of guanosine. However, it is not clear why modification of the O6 position of guanosine, which represents only a small fraction of the methylated DNA, is regarded as the major player in the drug’s cytotoxic action [7,37]. It is a well-known fact that in cancer cells, global demethylation is observed. It proceeds through oxidative demethylation, catalyzed by TET 1-3 oxo-reductases which require Fe^2+^ ion as a cofactor, a crucial catalyst of the Fenton reaction (Figure 2). At the same time, there is also a wealth of data showing that cancer cells show an elevated level of 8-hydroxydeoxyguanosine (8-oxo-dG), the oxidative DNA damage product. It is formed by the chemical reaction of guanosine residues with a hydroxyl radical (•OH) generated through the Fenton reaction [38]. Therefore, it is evident that oxidative stress contributes to increased damage, genome instability, and metastasis [39]. One can conclude that global DNA demethylation in cancer cells occurs spontaneously, and decreasing levels of m^5^C can be an appropriate marker of oxidative damage. Therefore, monitoring of the m^5^C and 8-oxo-dG levels at the same time in the DNA of cancer cells treated with different drugs such as juglone and its combination with TMZ will provide data for a better understanding of the mechanism of action of small molecular compounds.

## 2. Results

### 2.1. Cytotoxic Effect of Juglone and Temozolomide in T98G and HaCaT

The T98G and HaCaT cells were treated with increasing doses of juglone and TMZ in the range of 1–1000 and 1–2000 µM, respectively (Figure 3).

Juglone above 30 µM strongly inhibits the growth of both cells (Figure 3A), but the TMZ inhibitory effect is significantly weaker (Figure 3B). The viability of the cells at 2000 μM of TMZ is almost the same as at 40 μM of juglone (Figure 3B). One can notice that there are no significant differences in cytotoxicity affected by both agents in normal and cancer cells (Figure 3). Finally, the cell’s viability after juglone and TMZ treatment suggests that juglone is more toxic to the cell than TMZ. IC_50_ values calculated for juglone and TMZ are seen in Table 1.

Next, we analyzed the combined effect of juglone and TMZ on HaCaT and T98G cells (Figure 4). As one can see, normal cells (HaCaT) are less resistant to TMZ than T98G glioma cells, especially at high TMZ concentrations (Figure 4A,B). On the other hand, juglone at low concentrations shows a small effect on T98G cells (Figure 4). The addition of an increasing amount of TMZ potentiates the action of juglone (Figure 4). Finally, 100 μM of TMZ reduces T98G cell viability by ca 20% (Figure 4D). Low concentrations of juglone (1–30 μM) with a low amount of TMZ (30 μM) are less toxic than that induced with 100 μM (Figure 4). That suggests the synergistic effect of these drugs on cell viability [40,41].

### 2.2. Cell Cycle Analysis

T98G and HaCaT cell lines treated with juglone and TMZ alone or in combination were evaluated with a flow cytometer. In the absence of juglone, we did not observe any effects of TMZ on the HaCaT cell cycle (Figure 5A). On the other hand, juglone above the concentration of 5 µM with TMZ above 30 µM causes a significant decrease in the G1 phase. For the combined treatment of juglone and TMZ, the dominant effect of juglone is observed. One can also see a slight increase in the cells in the G2 and S phases (Figure 5A). For the T98G tumor cell line, 100 μM TMZ without juglone, a slight increase in the S phase and a decrease in G1 and G2 was observed (Figure 4B). In the case of juglone without TMZ, we observed a significant increase in the G1 phase and a decrease in the G2. Juglone treatment of T98G glioma cells in the presence of 1 µM TMZ results in a significant decrease in the G1 phase and an increase in phase S above 5 µM juglone concentration.

Furthermore, a high TMZ concentration (100 µM) and a juglone concentration of 25 µM cause big arrests of the cells in the G1 phase. One can suggest a sub-G1 phase population and cell death. The G2 phase disappears with a high TMZ concentration (Figure 5B). The number of cells in the S phase increased with the increase of up to 10 μM juglone concentration, and decreased at 25 µM juglone in the presence of 100 µM TMZ concentrations. This pattern is similar to that induced with 25 μM juglone without TMZ. These observations suggest that T98G cells are susceptible to the action of both juglone and TMZ.

### 2.3. The Effect of Juglone on Genomic DNA Methylation Level

The effect of juglone on HaCaT, T98G, U138, and U118 cell lines was analyzed on the global m^5^C amount in cellular DNA. Because the m^5^C level is a very sensitive marker, we used three different glioma cell lines and lower concentrations of drugs. The total DNA methylation level was checked after treatment with 0–150 µM of juglone and 0–100 µM of TMZ (Figure 6B). All glioma cell lines show a similar global methylation pattern which is different from normal cells. We noticed concentration- and time-dependent changes in m^5^C contents. Global DNA methylation of the normal cell line (HaCaT) did not change significantly in the presence of an increasing concentration of juglone (Figure 6B). The highest values of m^5^C in DNA are observed for the T98G cell line and the incubation time of 24 h (Figure 6A).

A longer incubation time (48 h) induces a decrease of the m^5^C content. This suggests that the demethylation process takes place. One can see that all doses of TMZ induce a lowering of the DNA (m^5^C) level, but juglone causes an apparent dose-dependent increase in genomic DNA methylation in the T98G cell line (Figure 6C). From that, it is clear that the effect of juglone and TMZ on the glioblastoma cell line depends on the concentration of both compounds.

### 2.4. Analysis of 8-oxo-dG and m^5^C Contents in the DNA of the T98G Cell Line Treated with Juglone and TMZ

A total amount of the 8-oxo-dG in genomic DNA from the T98G cell line was analyzed after treatment with juglone. It can be seen that juglone at high concentrations induces a high increase in the level of 8-oxo-dG in DNA. At the same time, the m^5^C amount was on the same level (Figure 7A). However, the combination of juglone and TMZ concentrations of 1, 30, and 100 µM induced changes in m^5^C and 8-oxo-dG synthesis. Hypomethylation for all concentrations of TMZ is evident (Figure 7B). On the other hand, we observed an increase of 8-oxo-dG for 1 µM TMZ. Further increase of TMZ concentration causes a decrease of 8-oxo-dG, particularly at 100 µM of TMZ, which was expected. The decrease in the amount of 8-oxo-dG suggests the induction of DNA repair mechanisms.

## 3. Discussion

Despite various available treatments and advances in chemotherapy, the prognosis for people diagnosed with glioblastoma is still poor. The current treatment includes maximal surgical resection, radiotherapy, and chemotherapy using TMZ [5]. TMZ is treating standard chemotherapy for GBM, but does not fulfil its role because many GBMs are resistant to its action. As such, there is a need to find a way to overcome these obstacles by developing new, more effective strategies for treating gliomas. Recently, much emphasis has been placed on bioactive small natural compounds isolated from plants that are not very toxic and may induce slight side effects. Here, we evaluated juglone’s effects on the viability of normal and cancer cell lines. To get an insight into the mode of juglone action, we analyzed global changes in m^5^C, the epigenetic marker, and 8-oxo-dG global oxidative damage marker levels. The properties and functions of both compounds are well known, and one can use them to monitor any cellular processes. Interestingly, juglone affects the level of both markers. The level of DNA m^5^C methylation was slightly increased, but the level of 8-oxo-dG was very high (Figure 7). This clearly suggests the toxicity of juglone at high concentrations. However, in combination with TMZ, juglone can help combat brain tumors. This is possible because the mode of action of these compounds is different. Juglone induces cytotoxicity by producing reactive oxygen species (ROS). The increasing level of ROS oxidizes proteins, lipids, and nucleic acids. DNA is one of the main components of the cell and a crucial target of various drugs. ROS oxidized DNA by reaction at the C-8 of guanosine with •OH, forming 8-hydroxy-deoxyguanosine (8-oxo-dG), which can be easily determined with HPLC/EC on the genomic scale [39]. 8-oxo-dG, as the hydroxyl radical DNA oxidation product, is also a good marker of global cellular damage. We found that juglone inhibits T98G cell’s progression from the G1 into the S phase and blocks cell proliferation. For normal cells, one can see an increased amount of cells in phases S and G2. The other mode of juglone action is a non-covalent intercalation into DNA. The C1-O side of juglone is embedded between two A-T base pairs of DNA, but the 5′-hydroxyl and 4-naphthoquinone groups extended to the outside of the DNA double helix [29]. Therefore, juglone can lock block DNA replication and transcription inhibition (G1 phase). One can speculate that this mechanism is very similar to that of berubicin, which also intercalates into DNA and interrupts topoisomerase II activity, resulting in the inhibition of DNA replication and repair [42]. Berubicin is proposed to be a good drug for GBM treatment due to its cability to cross the blood–brain barrier with the destination of a brain tumor [43]. Juglone is structurally similar to berubicin aglycone. Finally, juglone can also form DNA adducts and react with thiol groups through a Michael addition reaction. On the other hand, the mechanism of TMZ action on the cell is completely different from that of juglone. This is because they show various hydrophobicity evidenced by Log P, which is -2.8 and 1.92 for TMZ and juglone, respectively. TMZ methylates bases of DNA (m^7^G—70%; m^3^A—9%; O^6^mG—5%) but not cytosine at C5 (Figure 2) [37]. It is also known that components of DNA are oxidized due to indirect action of TMZ, which means that the level of m^5^C, an epigenetic marker in GBM cells, is reduced [44]. From our results, it is clear that both compounds, juglone, and TMZ, induce oxidative damage of guanosine and m^5^C residues in DNA. The combination of juglone with temozolomide modulates the level of m^5^C and 8-oxo-dG (Figure 8). As such, this composition clearly shows that epigenetic changes play the main role in juglone action, and suggest that juglone can be used for glioma treatment alone or in combination with temozolomide (Figure 8).

## 4. Materials and Methods

### 4.1. Chemicals and Reagents

Juglone and temozolomide (Merck, Darmstadt, Germany) were dissolved in dimethyl sulfoxide (DMSO, Sigma/Merck Darmstadt, Germany). [γ-^32^P] ATP (6000 Ci/mmol) was purchased from Hartmann Analytic GmbH (Braunschweig, Germany). T4 polynucleotide kinase was purchased from USB Thermo Fisher Scientific, Waltham, MA, USA. Micrococcal nuclease, spleen phosphodiesterase II, apyrase, P1 nuclease, thiazolyl blue tetrazolium bromide, inorganic salts, cellulose plates, and methanol were purchased from Merck (Darmstadt, Germany). A Genomic Mini kit for DNA isolation was supplied by A&A Biotechnology (Gdańsk, Poland).

### 4.2. Cell Line and Culture Conditions

Human glioblastoma (T98G, U138, U118) and human epidermal keratinocyte (HaCaT) cell lines were purchased from ATCC (Manassas, VA, USA). Both cell lines were cultured in EMEM medium from ATCC (USA). They were supplemented with 10% (*v*/*v*) fetal bovine serum (FBS, Sigma/Merck) and 10 mg/mL antibiotics (penicillin 100 U/mL 330 and streptomycin 100 μg/mL) from ATCC (Manassas, VA, USA). Cells were cultured at 37 °C with 5% CO_2_ in humidified air. After 24 h, cells were washed with phosphate-buffered saline (PBS, Merck), placed in a fresh medium, and treated with juglone and TMZ alone or with a mixture of juglone and TMZ.

### 4.3. Cell Viability/Proliferation Assay

Cell viability was evaluated with a dye-staining method, using 3-(4,5-dimethyl-2-thiazolyl)-2,5-diphenyl-2H-tetrazolium bromide (MTT) [45]. MTT was dissolved in Dulbecco’s phosphate buffered saline, pH = 7.4 (DPBS) to 5 mg/mL. Cells were seeded in 96-well culture plates at a density of 1 × 10^4^ cells/well and grown in the supplemented medium at 37 °C under a 5% CO_2_ atmosphere. The cells were treated with 1–2000 μM TMZ alone, or 1–1000 μM juglone alone, or 1–1000 μM juglone together with 1, 30, and 100 μM TMZ. Using combinations of juglone and TMZ at the concentrations mentioned above, we show the concomitant effect of that dual treatment. After 24 h, the supernatant was washed out, and 100 μL of MTT solution in medium (0.5 mg/mL final concentration of MTT) was added to each well for 2 h. During this time, cells were incubated at 37 °C under a 5% CO_2_ atmosphere. Viable cells with active metabolism convert MTT into a purple colored formazan product with an absorbance maximum near 570 nm. When cells die, they lose the ability to convert MTT into formazan, thus color formation serves as a useful and convenient marker of only the viable cells. After the incubation time, the unreacted dye was removed through aspiration. The formazan product of the MTT tetrazolium accumulates as an insoluble precipitate inside cells, as well as being deposited near the cell surface and in the culture medium. The formazan were dissolved in 100 μL/well DMSO by mixing 10 min prior to recording absorbance readings. The absorbance was measured spectrophotometrically in a multi-well Synergy2 plate reader (BioTek Instruments, Winooski, VT, USA) at a test wavelength of 570 nm and a reference wavelength of 650 nm. A signal is proportional to the number of viable cells present. The half-maximal inhibitory concentrations (IC_50_) were calculated by fitting experimental values to the sigmoidal bell-shaped equation using GraphPad Prism v5.01 (GraphPad Software, Inc., San Diego, CA, USA). The data obtained represent the means from four independent experiments.

### 4.4. Cell Lines’ Treatment with Juglone

Cells were seeded in 6-well culture plates at a density of 0.3 × 10^6^ cells/well and grown in the supplemented medium at 37 °C under a 5% CO_2_ atmosphere. Freshly prepared juglone stock solution was added directly to the culture medium (with 90–95% confluence) to get different 0.1–150 μM concentrations in DMSO and incubated for 3, 12, 24, and 48 h. For the control, the cells were treated with DMSO only. After 3–48 h of juglone treatment, the cell medium was aspirated, cells were washed with 500 μL of PBS, trypsinized by adding 200 μL of trypsin and incubation at 37 °C till the cells detached from the bottom of the plate wells, and collected by centrifugation at 4000 rpm for 10 min. The solution was discarded, and the cellular pellets were quickly frozen and stored at 20 °C for DNA isolation.

### 4.5. Cell Lines’ Treatment with the Combination of Juglone and Temozolamide

Cells were seeded in 6-well culture plates at a density of 0.3 × 10^6^ cells/well and grown in the supplemented medium at 37 °C under a 5% CO_2_ atmosphere. Freshly prepared juglone and TMZ stock solutions or dilutions were added directly to the culture medium to get the desired concentration. In experiments with TMZ, the final DMSO concentration in each cell culture was 0.8%. Cell cultures (with 90–95% confluence) were washed with 500 μL of PBS and placed in fresh medium and treated with 5–150 µM juglone by 24 h, or 5–150 µM juglone and 1, 30, 100 µM TMZ by 24 h. The control cells were treated with DMSO only. After the incubation time, the cell medium was aspirated, cells were washed with 500 μL PBS, trypsinized by adding 200 μL of trypsin and incubation in 37 °C till the cells detached the bottom of the plate wells, and collected by centrifugation at 4000 rpm for 10 min. Then, the solution was discarded, and the cellular pellets were quickly frozen and stored at 20 °C for DNA isolation.

### 4.6. Cell Cycle Analysis by Flow Cytometry

Next, 1.5 × 10^5^ cells were seeded onto 12-well cell culture plates and incubated for 24 h with 1–100 μM temozolomide and 0.5–150 µM juglone. Control cells were incubated in a fully supplemented growth medium without these compounds. At the end of incubation, the cells were harvested with trypsin and centrifuged at 200× *g* for 5 min, and the cell culture medium was removed. The pellet was washed twice with 1 mL of PBS, centrifuged as previously, and resuspended in 30 μL of PBS. Then, 1 mL of ice-cold 75% ethanol was added drop-by-drop with mixing and incubated for 2 h at −20 °C. Before the cell cycle analysis by flow cytometry was performed, fixed cells were stained with propidium iodide (PI). After centrifugation (200× *g* for 5 min) cells were washed twice with 1 mL of PBS and spun down. The pellet was incubated in 100 μL of PBS containing 1% of FBS, PI (5 μg), and RNase A (20 μg) for 30 min, at 37 °C in the dark. The PI fluorescence was measured by the FACS Calibur flow cytometer (Becton Dickinson, Franklin Lakes, NJ, USA). Data were analyzed by FlowJo software v.2012.

### 4.7. DNA Isolation from Cell Cultures

DNA from cells was extracted with the Genomic Mini kit according to the manufacturer’s instructions and incubated with RNase A. After centrifugation (15,000 rpm for 3 min), the supernatant was applied to a mini column and DNA bound to the column was eluted with Tris-buffer (pH 8.5) and stored at −20 °C for further analysis. The purity and concentration of DNA preparations was assessed by measuring UV absorbance measure at 260 and 280 nm. An A_260_/A_280_ ratio of 2.0–2.1 identified pure DNA.

### 4.8. Analysis of m^5^C Contents in DNA

The analysis of m^5^C was performed according to the given scheme (Figure 9).

DNA (dried, 1 μg) was dissolved in a succinate buffer (pH 6.0) containing 10 mM CaCl_2_ and digested with 0.001 units of spleen phosphodiesterase II and 0.02 units of micrococcal nuclease, in a total volume of 4 μL for 5 h at 37 °C. The DNA digest (0.2 μg) was labelled with 1 μCi [γ-^32^P]ATP (6000 Ci/mmol) and 1.5 units of T4 polynucleotide kinase in 10 mM bicine-NaOH buffer (pH 9.7), containing 10 mM MgCl_2_, 10 mM DTT, and 1 mM spermidine. Then, the apyrase was added to remove excess [γ-^32^P]ATP. The 3′nucleotide phosphate was cleaved off with 0.2 μg RNase P1 nuclease in 500 mM ammonium acetate buffer (pH 4.5). Identification of [γ-^32^P]m^5^C was performed with two-dimensional thin-layer chromatography (TLC) on cellulose plates using the solvent system isobutyric acid:NH_4_OH:H_2_O (66:1:17 *v*/*v*) in the first dimension, and 0.2 M sodium phosphate (pH 6.8)-ammonium sulfate-n-propyl alcohol (100 mL/60 g/2 mL) in the second dimension. Radioactivity was subsequently measured using a Fluoro Image Analyzer FLA-5100 with Multi Gauge 3.0 software (Fujifilm). Each analysis was repeated three times. For precise calculations, we evaluated spots corresponding not only to m^5^C, but also to products of its final degradation, such as cytosine (C) and thymine (T). The amount of m^5^C was calculated as R = [(m^5^C/m^5^C + C + T)] × 100 [32].

### 4.9. Analysis of 8-oxo-dG in DNA

DNA was dissolved in 200 µL of 40 mM sodium acetate buffer (pH 5.3) containing 0.1 mM ZnCl_2_, digested with P1 nuclease solution (30 µg), and incubated for 3 h at 37 °C. Then, 30 µL of 1M Tris–HCl pH 8.0 and 5 µL of alkaline phosphatase (1.5 units) solution was added, followed by 1 h incubation at 37 °C. DNA hydrolysate was purified using a cut-off of 10,000 Da filter units. The amount of 8-oxo-dG in DNA was determined using HPLC (Agilent Technologies 1260 Infinity, CA, USA) with two detectors working in series: the 1260 Diode Array Detector and the Coulochem III Electrochemical Detector (ESA Inc., Chelmsford, MA, USA). Isocratic chromatography of DNA hydrolysate was performed using a solution of 50 mM CH_3_COONH_4_ at pH 5.3 and methanol (93:7). Analysis of dG (for reference) was performed at 260 nm. The amount of 8-oxo-dG was determined with the following electrochemical detection settings: guard cell +400 mV, detector 1: +130 mV (screening electrode), detector 2: +350 mV (measuring electrode set on the 100 nA sensitivity) [39].

### 4.10. Calculation of the Total Amount of m^5^C and 8-oxo-dG in Human DNA

The number of modified bases in DNA was calculated on the basis of global genome composition—3.05 × 10^9^ bases (100%), where C—624 × 10^6^ (20.5%), T—905 × 10^6^ (29.6%), G—623 × 10^6^ (20.4%), A—901 × 10^6^ (29.5%), and m^5^C—31 × 10^6^ (1%). The amount of m^5^C (%) in pyrimidines in DNA was determined from TLC analysis with the formula R [%] = m^5^C × 100/(C + T). The total number of m^5^C in the genome was calculated from the formula m^5^C = (1 498 333 975) × R/100. The input amount of guanosine was necessary to determine 8-oxo-dG contents. It was calculated from diode array detector (PAD) measurements using the Avogadro number N_G_ = 6.02 × 10^20^ × b(mAU)/a(mAU) standard. The 8-oxo-dG nucleoside amount was estimated with the electrochemical detector N_8-oxo-dG_ = 6.02 × 10^20^ × d(nA)/c(nA) standard. The total number of 8-oxo-dG = 623 × 10^6^ × N_8-oxo-dG_/N_G_ [46].

### 4.11. Statistical Analysis

Microsoft Excel 2010 software with a data analysis package was used for the statistical analysis of all data. The data are the result of three independent experiments. The descriptive statistics function generated the mean and SD, and the results were expressed in the error bars. One-tailed *t*-test was used to calculate significant differences in m^5^C content for tested samples compared to control experiments, and a *p*-value < 0.05 was considered as significant.

## Figures and Tables

**Figure 1 ijms-24-06998-f001:**
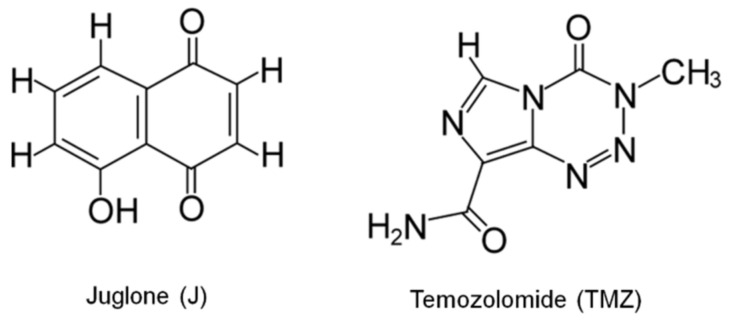
Chemical formulae of juglone (Mw 174,16) and temozolamide (Mw 194,15).

**Figure 2 ijms-24-06998-f002:**
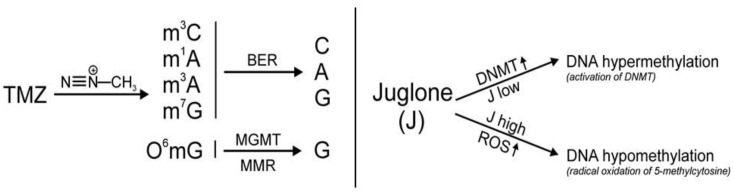
Mechanism of action of temozolomideTMZ and juglone BER-Base Excision Repair, MGMT-Methylguanine–DNA methyltransferase, MMR-Mismach Repair, DNMT-DNA methyl transferase, ROS-Reactive Oxygen Species.

**Figure 3 ijms-24-06998-f003:**
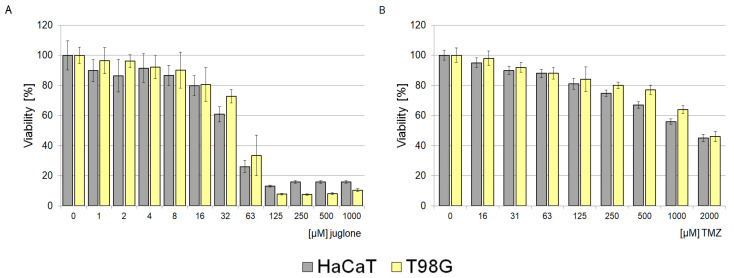
Cytotoxicity of juglone (**A**) and TMZ (**B**) effect on human keratinocyte (HaCaT), glioblastoma (T98G) cell lines. The cells were treated with 1–1000 μM of juglone and 1–2000 μM of TMZ for 24 h. Cell viability was determined using the MTT method.

**Figure 4 ijms-24-06998-f004:**
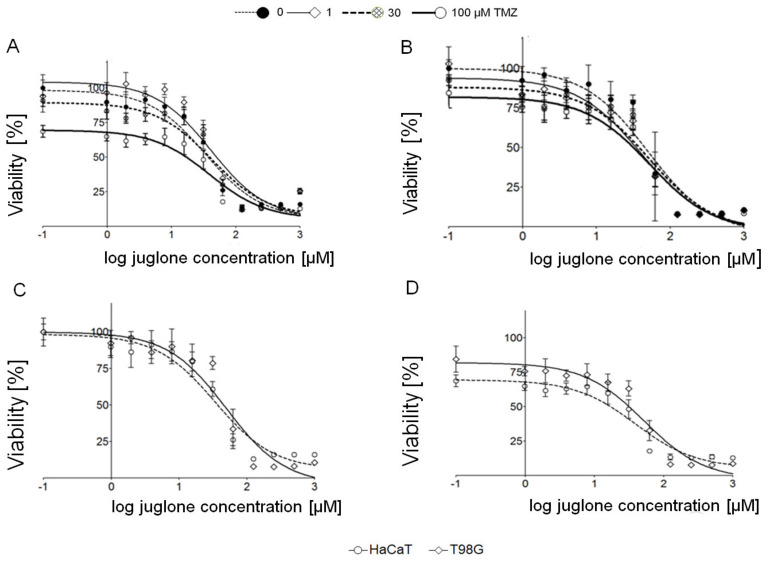
Juglone effect on (**A**) human keratinocyte (HaCaT) and (**B**) glioblastoma (T98G) cell lines alone and with TMZ. Cells proliferation was analyzed 24 h after supplementation with juglone (1–1000 μM, −1 to 3 in logarithmic scale) and TMZ (0, 1, 30, 100 μM). (**C**) Juglone concentration 1–1000 μM (−1 to 3 in logarithmic scale) without TMZ and (**D**) with 100 μM TMZ.

**Figure 5 ijms-24-06998-f005:**
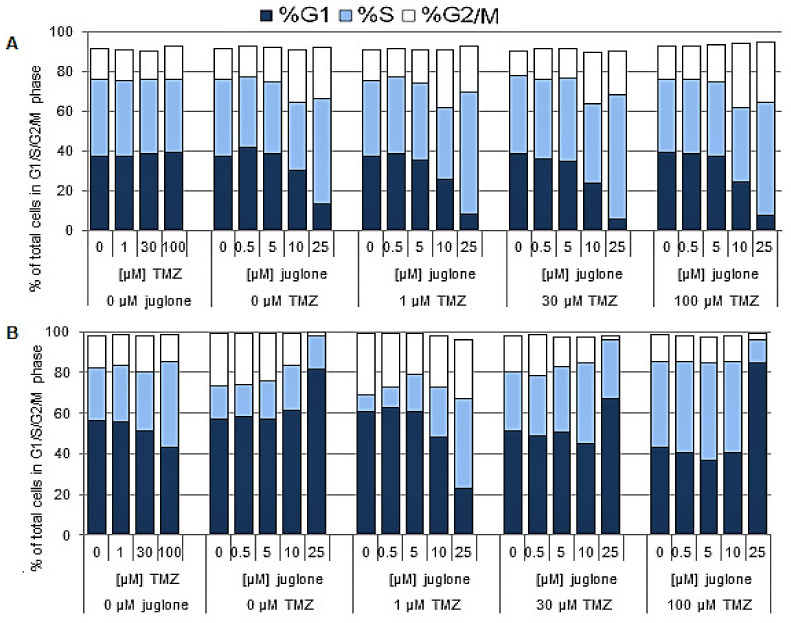
The effect of TMZ and juglone on the HaCaT (**A**) and T98G (**B**) cell cycle. The percentage of cells in each phase of the cell cycle is showed. All experiments were performed in triplicate.

**Figure 6 ijms-24-06998-f006:**
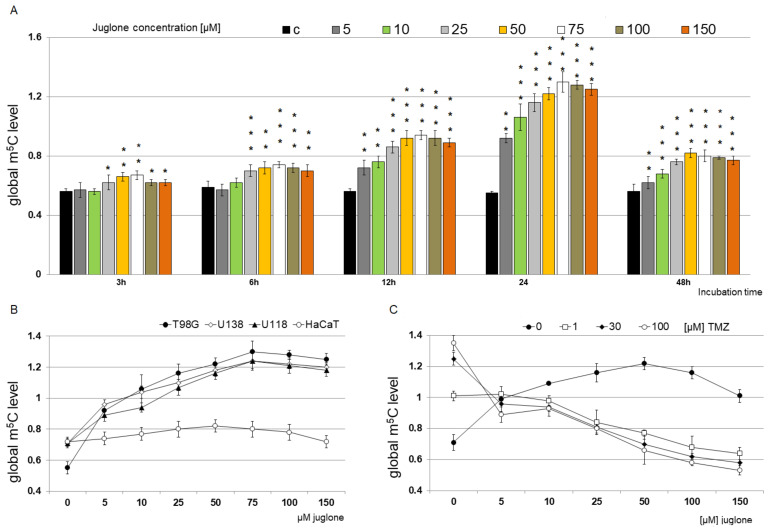
The effect of juglone on the total DNA (m^5^C) methylation level of the T98G cell line (**A**). The analysis was performed after 3, 6, 12, 24, and 48 hrs of incubation at different juglone concentrations (0–150 μM). Control cells (c) were treated with DMSO only. The m^5^C values are means from three experiments ± SD. Asterisks indicate a significant difference (* *p* < 0.05, ** *p* < 0.01, *** *p* < 0.001) from the control (DMSO) value. A very small DNA hypermethylating effect is observed (for incubation times 3, 6, and 48 h), and clear hypomethylation is seen for incubation times of 12 and 24 h. (**B**) The effect of juglone on DNA methylation in different cell lines, T98G, U138, U118, and HaCaT (time incubation was 24 h). (**C**) The effect of the action of juglone/TMZ on the total DNA methylation level in the T98G cell line. The analysis was performed after 24 h of incubation at a given juglone concentration (0–150 μM) and TMZ concentration (0–100 μM).

**Figure 7 ijms-24-06998-f007:**
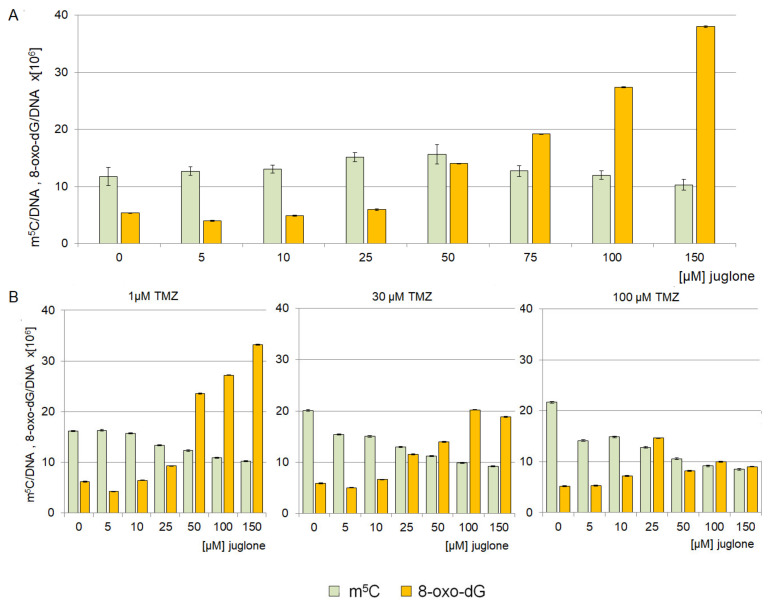
Contents of m^5^C and 8-oxo-dG in DNA from T98G cell line after treatment with juglone alone (**A**) and in combination with TMZ (**B**). The analysis was performed after 24 h of incubation. Control experiment cells were treated with DMSO only. The R values are the means from four experiments.

**Figure 8 ijms-24-06998-f008:**
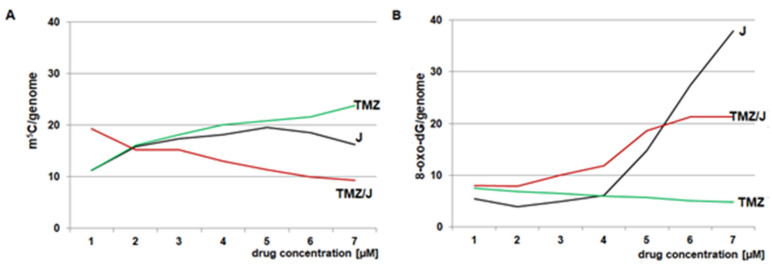
The juglone and TMZ alone or in combination affects DNA (m^5^C) methylation (**A**) and DNA (8-oxo-dG) oxidation (**B**). The concomitant application of both drugs induces DNA hypomethylation and hyper DNA oxidation. Drug concentration: 0, 0.5, 10, 25, 50, 100, 150 μM for 1–7 concentration points, respectively.

**Figure 9 ijms-24-06998-f009:**
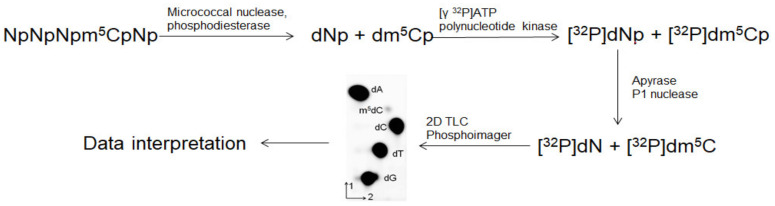
Analysis of m^5^C in DNA.

**Table 1 ijms-24-06998-t001:** IC_50_ values of juglone and TMZ and for T98G and HaCaT cells at 24 h treatment.

	HaCaT	T98G
Juglone	34.0 μM	49.6 μM
Temozolomide	420 μM	1430 μM

## Data Availability

The data presented in this work are available upon request from the corresponding author.

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
