# Peer review of "Juglone in Combination with Temozolomide Shows a Promising Epigenetic Therapeutic Effect on the Glioblastoma Cell Line"

_ijms, 2023, doi:10.3390/ijms24086998_

Round 1

Reviewer 1 Report

The presented paper covers an important topic of the search for new therapeutics to treat GBM. Nevertheless, it requires a significant amount of work and I'd recommend a major review.

1) Images quality is very low and makes images almost impossible to read. Why image #7 is presented in color and the other ones in grayscale?

2) Presented chemical structures have the wrong length of bonds that should be adjusted, e.g. the double bond is shorter than the single bond.

3) Size of the font in images needs to be consistent and readable.

4) I'd recommend adding to the introduction the image presenting a mode of action for TMZ and juglone. Additionally, discuss their mode of action in the intro text.

5) The mechanism of resistance to TMZ is known and should be discussed.

6) The text misses direct sentences showing the impact of this research and motivation. It's mentioned in the discussion but should be clearly presented in the intro.

7) Text has some formatting issues, e.g. additional space line #90, use of "," instead of "." in numeric values, referring to figures "Figure X", "Fig. X". I'd recommend using active voice in data presentation.

8) MTT method is considered not the best choice for many drugs since they may change the potential of the membrane and change the uptake profile. I'd recommend a newer method that monitors the disruption/permeability of the membrane and the activity of mitochondria at the same time.

9) The discussed changes in toxicity (Fig#3) are not easy to read from the graphs, looks like there's no significant difference between C and D.

10) Fig#5, graphs need the legend on the axis

11) Authors in the middle of the research introduced additional cell lines but the purpose of this action is not explained. The rationale behind this action should be provided.

12) Analyzing the concentration of m5C and 8oxoG authors changed the range of drug concentration, it should be discussed why.

13) In the discussion section, I'd recommend an additional image showing the mechanisms/models and modes of action suggested by the authors.

14) In M&M, authors describe that juglone was added to cells presenting 90-95% confluency. Then they were cultured for the next 2 days, it may result in a significant overgrowth of cells, additional stress, and a change in the response of cells to the drug.

Reviewer 2 Report

The authors investigated the therapeutic effect of juglone on GBM cell models. The paper is interesting but not extremely innovative.  The design of the manuscript is quite clear but the methods used by the authors should be better described. Results should be improved to adequately support their final conclusions. I have some major and minor concerns that should be addressed to achieve publication priority.

Major points:

1. Abstract and result section should be better described. They are not so informative.

2. Material and methods section should be described in deeper details, including statistica

analysis.

3. The authors should test juglone citotoxicity on more than  one GBM cell lines

4. They should perform RNAseq to better understand the mechanism of action to Juglone. Otherwise they should evaluate some specific targets (DNMTs, ecc) and study the effects on oxidative stress

6.in vivo experiments in orthotopic mice are needed to study the ability of juglone to cross the bbb and induce anti tumor effects

Minor points: 

1.Some errors throughout the text.

2.Figure legend lack some important information and they should be completely and accurately rewritten.

Round 2

Reviewer 1 Report

The quality of images #3, 5, and 7 is very bad and needs to be corrected. I'd recommend exporting high-quality images (at least 300 dpi).
